# Glycolytic reprograming in *Salmonella* counters NOX2-mediated dissipation of ΔpH

Sangeeta Chakraborty[1], Lin Liu [1], Liam Fitzsimmons[1], Steffen Porwollik[2], Ju-Sim Kim[1], Prerak Desai [2], Michael McClelland [2] & Andres Vazquez-Torres[1,3 ✉]

The microbial adaptations to the respiratory burst remain poorly understood, and establishing how the NADPH oxidase (NOX2) kills microbes has proven elusive. Here we demonstrate that NOX2 collapses the ΔpH of intracellular *Salmonella* Typhimurium. The depolarization experienced by *Salmonella* undergoing oxidative stress impairs folding of periplasmic proteins. Depolarization in respiring *Salmonella* mediates intense bactericidal activity of reactive oxygen species (ROS). *Salmonella* adapts to the challenges oxidative stress imposes on membrane bioenergetics by shifting redox balance to glycolysis and fermentation, thereby diminishing electron flow through the membrane, meeting energetic requirements and anaplerotically generating tricarboxylic acid intermediates. By diverting electrons away from the respiratory chain, glycolysis also enables thiol/disulfide exchange-mediated folding of bacterial cell envelope proteins during periods of oxidative stress. Thus, primordial metabolic pathways, already present in bacteria before aerobic respiration evolved, offer a solution to the stress ROS exert on molecular targets at the bacterial cell envelope.

---

[1] Department of Immunology & Microbiology, University of Colorado School of Medicine, 12800 E. 19th Ave, Mail Box 8333, Aurora, CO 80045, USA. [2] Department of Microbiology and Molecular Genetics, University of California Irvine School of Medicine, 240 Med Sci Bldg., Irvine, CA 92697, USA. [3] Veterans Affairs Eastern Colorado Health Care System, Denver, CO, USA. ✉email: andres.vazquez-torres@cuanschutz.edu

 

ROS generated through the enzymatic activity of the phagocyte NADPH oxidase (NOX2) are among the most potent host defenses pathogenic microorganisms face during infection. The significance of the respiratory burst in host immunity is illustrated by the recurrence of opportunistic *Staphylococcus aureus*, *Aspergillus fumigatus* or *Salmonella enterica* infections in patients carrying mutations in membrane-bound or cytosolic subunits of the NOX2 complex[1]. Despite its paramount role in host defense, our understanding of the molecular mechanisms by which NOX2 exerts its powerful antimicrobial activity remains a subject of contention[2–5]. It is widely accepted that reactive oxygen species generated in the respiratory burst damage DNA via Fenton-mediated chemistry and oxidize protein cysteine residues and metal cofactors in regulatory and metabolic proteins[6–8]. However, others have argued that phagocytic superoxide primarily damages extracytoplasmic targets[9]. How the bacterium mitigates this attack is also only partly understood[4,6,7,9].

The following investigations demonstrate that the profound antimicrobial activity of NOX2 against intracellular *Salmonella* involves inhibition of ΔpH across the membrane. In turn, *Salmonella* adapts to the challenge oxidative stress imposes to membrane energetics by shifting redox balance from oxidative phosphorylation to glycolysis.

## Results

**Glycolysis promotes resistance to $H_2O_2$ killing**. To gain insights into the mechanisms of oxidative killing and the adaptations that protect *Salmonella* against oxidative stress, a library of 140,000 Tn5 mutants of *Salmonella* serovar Typhimurium strain 14,028 s was exposed to $H_2O_2$. Fitness analysis confirmed the importance of divalent cation transporters *corA* and *mgtB*, glutathione synthase *gshB*, and the two-component regulatory system *phoP-phoQ* in the resistance of *Salmonella* to oxidative stress (Supplementary Data 1 and 2). Our analysis also found roles for phosphoglycerate mutase *gpmA* (Fig. 1a) and pyruvate kinase *pykF* (Supplementary Data 2) in resistance to $H_2O_2$ killing. Invariably, $H_2O_2$ exerted its most negative effects against *gpmA* transposon mutants, regardless of whether the libraries were grown in glucose and/or Casamino acids as sole carbon and energy sources (Supplementary Data 2). Defined deletions confirmed that manganese-independent *gpmA* phosphoglycerate mutase (Fig. 1b, Supplementary Figs. 1a,b), not the manganese-dependent *gpmB* isoform (Supplementary Fig. 1b), protects *Salmonella* against $H_2O_2$ killing. Metabolism of manganese appears to be under pressure in cells undergoing oxidative stress, as suggested by the negative selection exerted by $H_2O_2$ against *mntH* Tn mutants deficient in the high affinity manganese uptake system (Fig. 1a, Supplementary Data 2). Thus, the preferential utilization of manganese-independent phosphoglycerate mutase by *Salmonella* undergoing oxidative stress might reflect limitations for manganese that follow exposure to $H_2O_2$.

The role played by *gpmA* in resistance to $H_2O_2$ relies on its phosphoglycerate mutase enzymatic activity that isomerizes 3−phosphoglycerate to 2-phosphoglycerate, because a strain expressing the catalytically inactive *gpmA* R10A H11A variant was readily killed by $H_2O_2$ (Fig. 1b). The hypersusceptibility of Δ*gpmA Salmonella* to $H_2O_2$ was not evident when the bacteria were grown on 2-phosphoglycerate, pyruvate, acetate, fumarate or malate, which enter metabolism below 3−phosphoglycerate (Fig. 1c, Supplementary Fig. 2). Succinate increased resistance of Δ*gpmA Salmonella* to $H_2O_2$, but did not reach the levels of wild-type controls (Fig. 1c). It is unclear why succinate did not fully recover resistance of Δ*gpmA Salmonella* to $H_2O_2$ but, as it will be discussed later, it is possible that the electrons shuttled into the

ETC in the utilization of succinate may have contributed to the partial resistance provided by succinate. The addition of the iron chelators deferoxamine or dipyridyl did not prevent $H_2O_2$ killing of Δ*gpmA Salmonella* (Supplementary Fig. 1c). Moreover, Δ*gpmA Salmonella* exhibited comparable catalase activity to wild-type controls (Supplementary Fig. 1d). Together, these data rule out genotoxicity or defective $H_2O_2$ consumption as the underlying causes for the increased sensitivity of *Salmonella* deficient in glycolysis to oxidative stress.

**Glycolysis protects *Salmonella* against NOX2 killing**. We found that Δ*gpmA Salmonella* are attenuated in immunocompetent C57BL/6 mice (Fig. 1d, Supplementary Fig. 1e), a phenotype that could be complemented by a wild-type but not the catalytically deficient R10A H11A *gpmA* allele (Supplementary Fig. 1c). These findings are consistent with the importance glucose utilization and glycolysis play in *Salmonella* pathogenesis[10]. Remarkably, Δ*gpmA Salmonella* became virulent in $nox2^{-/-}$ mice lacking the gp91*phox*-encoded membrane subunit of the NADPH oxidase (Fig. 1d). A Δ*pfkAB* mutant deficient in both isoforms of the glycolytic enzyme phosphofructokinase also became virulent in $nox2^{-/-}$ mice (Fig. 1d), which contrasts with the profound attenuation of this mutant in $nos2^{-/-}$ mice deficient in inducible nitric oxide synthase[11]. Collectively, these data indicate that glycolysis plays a previously unknown role in the resistance of *Salmonella* to NOX2, and suggest that utilization of glucose in preference to 50 host-derived nutrients accessible during infection cannot simply be explained by availabilty[12] but rather by a hitherto unknown role of glycolysis in resistance to oxidative stress. The role played by glycolysis in resistance to oxidative stress is surprising if one considers that the oxidation of glyceraldehyde 3-phosphate dehydrogenase (GAPDH), a step just upstream of GpmA, shifts reducing power into the pentose phosphate pathway for synthesis of NADPH that fuels glutathione- and thioredoxin-dependent antioxidant defenses[13–15].

**Electron flow through the membrane promotes $H_2O_2$ killing**. $H_2O_2$ exhibited robust antimicrobial activity against aerobically grown Δ*gpmA Salmonella*, but negligible killing against anaerobic cultures (Fig. 1e). The resistance of anaerobic *Salmonella* to $H_2O_2$ is unexpected considering that the reducing cytoplasm of anaerobic bacteria is loaded with $Fe^{2+}$ readily available to catalyze Fenton-mediated genotoxicity. Nonetheless, $H_2O_2$ mediated substantial killing of anaerobic *Salmonella* grown in the presence of the terminal electron acceptor $NO_3^-$ (Fig. 1f), particularly in bacteria with diminished glycolytic activity ($p < 0.01$ when Δ*gpmA Salmonella* is compared to wild-type controls). *Salmonella* deficient in *nuo-* or *ndh*-enocded NADH dehydrogenases were resistant to $H_2O_2$ when tested under anaerobic conditions in the presence of $NO_3^-$ (Supplementary Fig. 1f). Together these findings strongly suggest that flow of electrons through the respiratory chain, rather than reactive oxygen species produced endogenously by the univalent reduction of $O_2$, potentiates the antimicrobial activity of $H_2O_2$.

**Fermentation and reductive TCA boost resistance to NOX2**. To gain further insights into the mechanisms by which glycolysis increases resistance to oxidative stress, we analyzed glycolytic and tricarboxylic acid cycle (TCA) metabolites of control and $H_2O_2$-treated *Salmonella*. Intermediates of glycolysis and the oxidative branch of the TCA, as well as lactate, acetyl phosphate and succinate were enriched in *Salmonella* treated with $H_2O_2$ (Fig. 2a, Supplementary Fig. 3 and Supplementary Data 3 and 4). Similar to wild-type controls, $H_2O_2$-treated Δ*gpmA Salmonella* accumulated oxidative TCA products, but failed to accumulate lactate

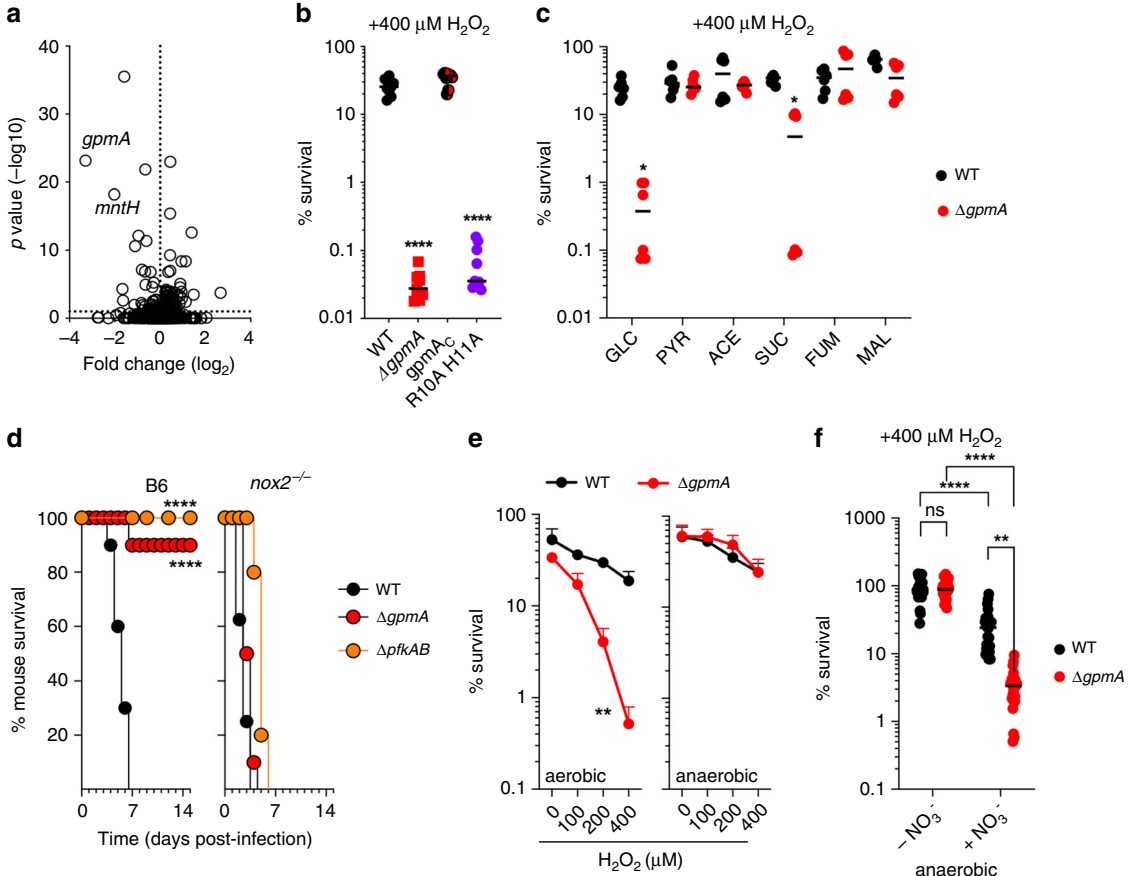

**Fig. 1 Glycolysis protects *Salmonella* against oxidative stress engendered by NOX2. a** Tn5 mutants differentially selected after 2 h of treatment with 2.5 mM $H_2O_2$, compared to untreated controls aerobically grown in MOPS-glucose. The specimens in these experiments were treated at $OD_{600}$ of 0.6 and a concentration of $7$-$8 \times 10^8$ CFU/ml. $N = 5$. **b** Survival of $2 \times 10^5$ CFU of *Salmonella* after 2 h of 400 µM $H_2O_2$ treatment in MOPS-glucose media. The $\Delta gpmA$ mutant was complemented with wild-type (gmpAc) or catalytically inactive (R10A H11A) *gpmA* alleles. Mean ± SD; $N = 9$. **c** Wild-type and $\Delta gpmA$ *Salmonella* were grown to $O.D_{600}$ 0.4 in MOPS media supplemented with glucose (GLC), pyruvate (PYR), acetate (ACE), succinate (SUC), fumarate (FUM) or malate (MAL). Where indicated, the specimens were treated for 2 h with 400 µM $H_2O_2$. Mean ± SD; $N = 6$. **d** C57BL/6 and congenic $nox2^{-/-}$ mice were inoculated i.p. with ~200 CFU of the indicated *Salmonella* strains. Mean ± SD; $N = 10$ except 9 for $nox2^{-/-}$ mice infected with wild-type *Salmonella*. **e** Killing of *Salmonella* 2 h after treatment with increasing concentrations of $H_2O_2$. Where indicated, the bacteria were grown anaerobically for 24 h before challenge with $H_2O_2$. Mean ± SD; $N = 5$ for aerobic cultures, $N = 6$ for anaerobic cultures. **f** Effect of 50 mM sodium nitrate on the susceptibility of anaerobic *Salmonella* exposed to 400 µM $H_2O_2$ in MOPS-glucose media for 2 h. Mean ± SD; $N = 27$. The data were analyzed by one-way (**b**) or two-way (**c**, **f**) ANOVA, log-rank Mantel-Cox test (df = 2) (**d**) or paired, one tail *t*-test (**e**). \*, \*\*, \*\*\*, \*\*\*\**p* < 0.05, 0.01, 0.001, 0.0001, respectively. All measurements were taken from distinct samples. Source data are included in Source Data file.

(Fig. 2b, Supplementary Fig. 3). In addition, untreated $\Delta gpmA$ *Salmonella* had increased basal concentrations of 3-phosphoglycerate, phosphoenolpyruvate (PEP), and acetyl CoA, but lower levels of malate and fumarate (Fig. 2b and Supplementary Fig. 3). $H_2O_2$ treatment did not increase production of succinate in $\Delta gpmA$ *Salmonella*. No differences in GSH ($p = 0.23$) and GSSG ($p = 0.12$) were found between untreated or $H_2O_2$-treated wild-type and $\Delta gpmA$ *Salmonella* (Supplementary Fig. 4). Collectively, these findings indicate that *Salmonella* undergoing oxidative stress favor glycolysis to balance NADH/NAD$^+$ and generate ATP. Moreover, *Salmonella* generate oxaloacetate by the anaplerotic use of PEP to reconstitute TCA intermediates for anabolism.

We tested whether anaplerotic utilization of PEP, as well as acetate and lactate fermentation, contribute to the adaption of *Salmonella* to oxidative stress. In contrast to an *ldhA* mutant deficient in lactate dehydrogenase, isogenic controls deficient in *ackA pta*, encoding acetate kinase and phosphotransacetylase, and to a lesser extent *ppc*, encoding PEP carboxylase, showed increased susceptibility to $H_2O_2$ (Fig. 2c). Given their

susceptibility to $H_2O_2$, we were surprised to find that strains of *Salmonella* bearing mutations in $\Delta ackA$ $\Delta pta$ or $\Delta ppc$ killed C57BL/6 mice in an acute model of infection that is dominated by the antimicrobial activity of NOX2 (Supplementary Fig. 5a). Nonetheless, a strain with mutations in all three *ackA pta* and *ppc* genes was avirulent in C57BL/6 mice, but killed $nox2^{-/-}$ mice unable to sustain a respiratory burst (Fig. 2d). The antioxidant function of *ppc* could reflect the splitting of oxaloacetate into oxidative or reductive TCA. To shed light into this dichotomy, $\Delta ackA$ $\Delta pta$ alleles were combined with mutations in *gltA* or *fumB*, genes that encode for citrate synthase or fermentative fumarase, respectively. These investigations showed that $\Delta ackA$ $\Delta pta$ $\Delta gltA$ and $\Delta ackA$ $\Delta pta$ $\Delta fumB$ *Salmonella* strains, but not the single $\Delta gltA$ and $\Delta fumB$ isogenic controls (Supplementary Fig. 5b), were attenuated in NOX2-expressing mice (Fig. 2e). Therefore, in its adaptation to the oxidative stress generated in the innate response, *Salmonella* exploit glycolysis for energetic and redox balance as well as for anaplerotic replenishment of TCA intermediates from PEP.

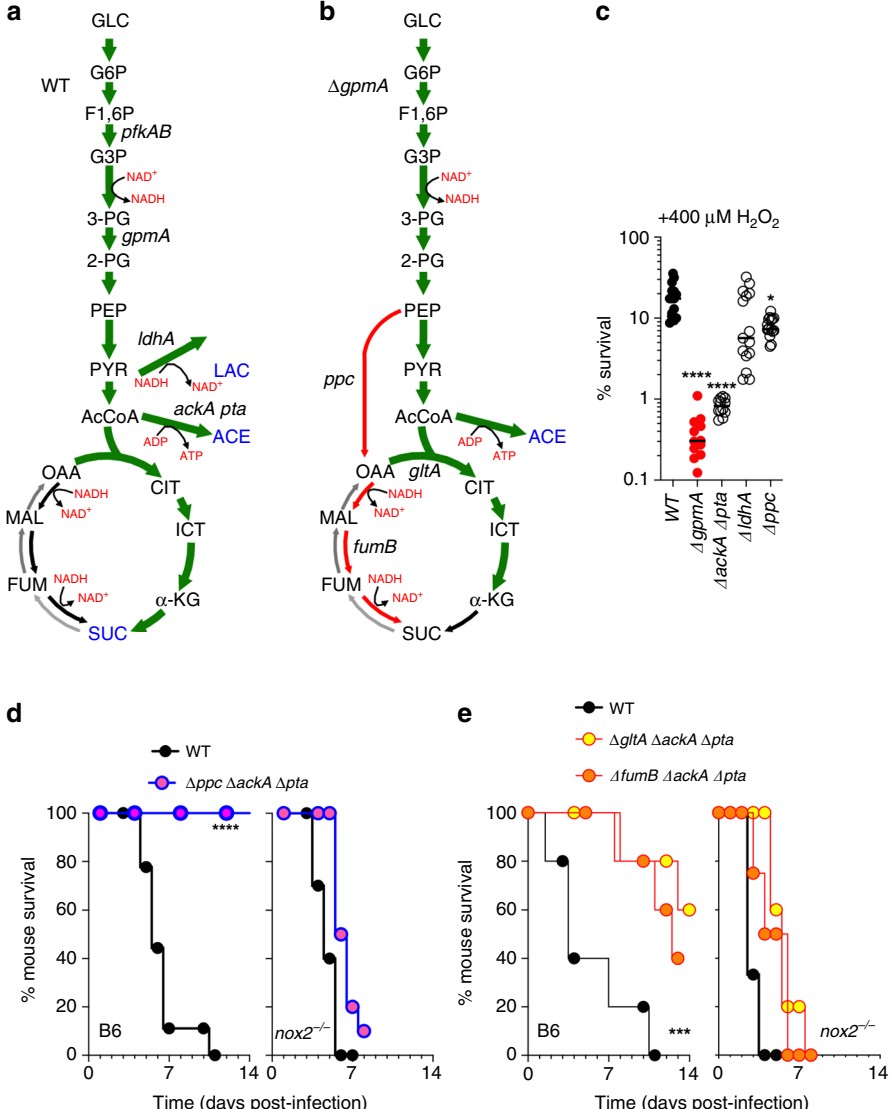

**Fig. 2 Oxidative stress induces adaptive glycolytic switch.** Schematic representation of central metabolites in **a** wild-type and **b** Δ*gpmA Salmonella* after growth for 30 min with 2.5 mM $H_2O_2$ in aerobic MOPS-glucose media. Green arrows denote pathways upregulated in response to $H_2O_2$. In blue is shown increased production of mixed fermentation products lactate (LAC), acetate (ACE) and succinate (SUC). Red arrows denote defective anaplerotic utilization of phosphoenolpyruvate (PEP), which accumulates in Δ*gpmA Salmonella*, for the formation of reductive TCA intermediates, which are present in lower concentrations in Δ*gpmA Salmonella* (see Supplementary Fig. 3 and Supplementary Data 3 and 4). Glucose (GLC), glucose 6-P (G6P), glyceraldehyde 3P (G3P), 3-phosphoglycerate (3-PG), 2-phosphoglycerate (2-PG), pyruvate (PYR), acetyl-CoA (AcCoA), citratre (CIT), isocitrate (ICT), α-ketoglutarate (α-KT), succinate (SUC), fumarate (FUM), malate (MAL), oxaloacetate (OAA). Phosphofructokinase (*pfkAB*), phosphoglycerate mutase (*gpmA*), lactate dehydrogenase (*ldhA*), PEP carboxylase (*ppc*), citrate synthase (*gtlA*), fumarase B (*fumB*), acetate kinase (*ackA*), phosphotranacetylase (*pta*). $N = 5$. **c** Killing of *Salmonella* 2 h after exposure to 400 μM $H_2O_2$ in MOPS-glucose media. Median; $N = 15$ except for Δ*gpmA* and Δ*ldhA* where $N = 12$. The data were analyzed by one-way ANOVA. *, ****$p < 0.05$, 0.0001, respectively, df = 59, F = 19.34. **d, e** Survival of C57BL/6 and $nox2^{-/-}$ mice after i.p. challenge with ~200 CFU of *Salmonella*. Data in d are $N = 9$ for C57BL/6 and 10 for $nox2^{-/-}$ mice; Data in e are $N = 5$. *, **, ***, $p < 0.05$, 0.01 and 0.001, respectively, by Log-rank Mantel-Cox test (d, df = 1, e, df = 2)(see also Supplementary Figs. 5a, b). All measurements were taken from distinct samples. Source data are included in Source Data file.

**$H_2O_2$ diminishes ΔpH in *Salmonella*.** GAPDH oxidizes glyceraldehyde-3P in glycolysis. In the process, NAD$^+$ accepts the hydrogen from the aldehyde group of glyceraldehyde-3-P to generate NADH. To ensure sustained GAPDH-dependent glycolytic activity, NAD$^+$ must be regenerated. Facultative anaerobes replenish NAD$^+$ from NADH by fermenting glucose to lactate, ethanol, and succinate. Given the defective production of succinate and lactate by Δ*gpmA Salmonella*, we compared NADH/NAD$^+$ ratios in wild-type and Δ*gpmA Salmonella*. We found that Δ*gpmA Salmonella* contained lower NADH/NAD$^+$ ratios (Fig. 3a) and slightly higher ATP concentrations

(Supplementary Fig. 6a) than wild-type controls, demonstrating that Δ*gpmA Salmonella* are at least as capable of balancing redox and producing ATP as wild-type cells. Because Δ*gpmA Salmonella* is defective in NADH-consuming fermentation reactions, we measured the capacity of this mutant to aerobically respire, an activity that oxidizes NADH. Polarographic studies showed that Δ*gpmA Salmonella* consumes more $O_2$ than wild-type controls, but showed similar rates as wild-type controls when complemented with a *gpmA* allele expressed from the low copy pWSK29 plasmid (Fig. 3b). Enhanced respiratory activity provides a reasonable explanation for the ability of

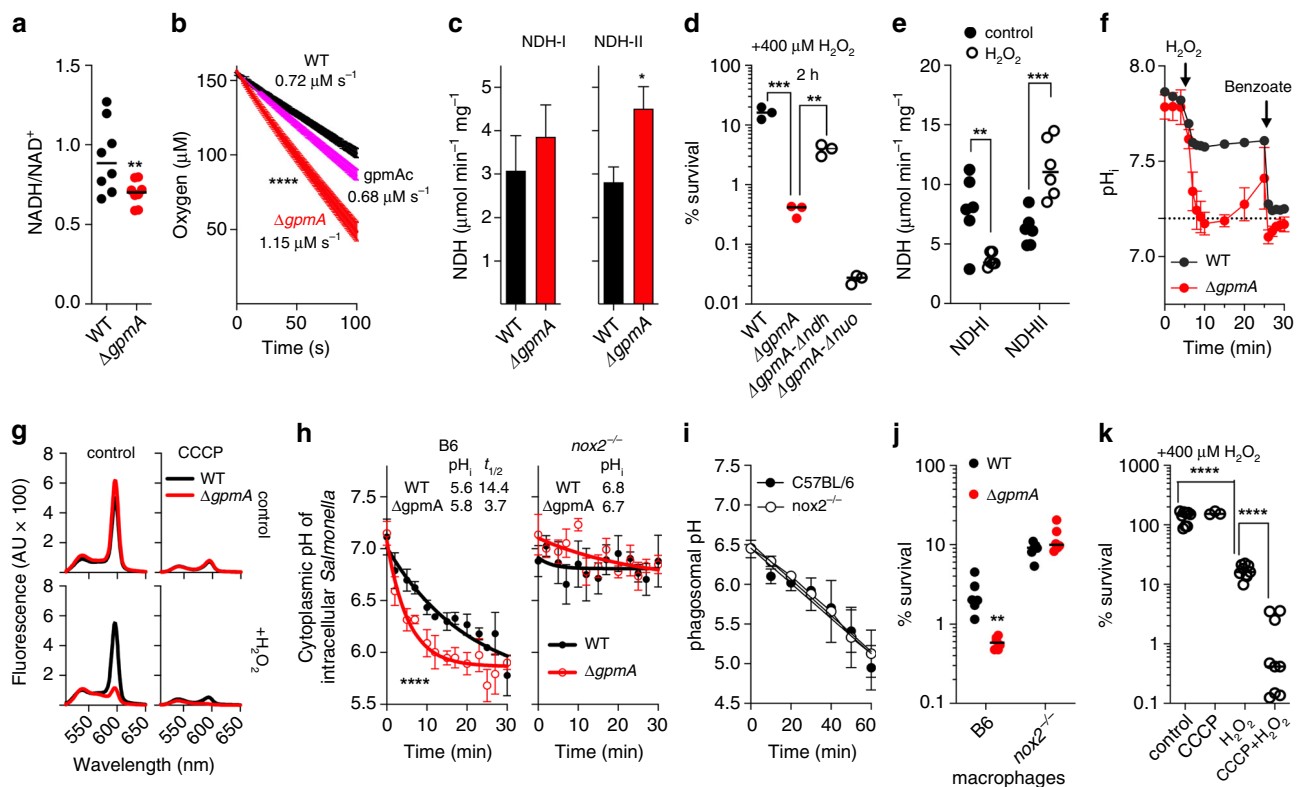

**Fig. 3 Glycolysis preserves ΔpH and mediates antioxidant defense. a** $NAD^+$ and NADH in log phase wild-type (WT) or Δ*gpmA Salmonella* grown in aerobic MOPS-glucose media. Mean ± SD; $N = 8$. **b** Aerobic respiration of log phase *Salmonella* grown in MOPS-glucose media was measured by polarography. Mean ± SEM; $N = 6$ except 4 for the complemented strain. **c** NADH-I and II dehydrogenase enzymatic activities of log phase *Salmonella* in MOPS-glucose media. Mean ± SD; $N = 12$. **d** Susceptibility of log phase *Salmonella* in MOPS-glucose media to 400 μM $H_2O_2$. Mean ± SD; $N = 3$. **e** NADH dehydrogenase enzymatic activity was determined in membranes of aerobic *Salmonella* in LB. Some cells were treated with 50 μM $H_2O_2$ for 10 min. Mean ± SD; $N = 6$. **f** Intracellular pH ($pH_i$) of log phase *Salmonella* grown aerobically in MOPS-glucose media, pH 7.2, after treatment with 400 μM $H_2O_2$ as measured ratiometrically with the pHluorin. When indicated, the ΔpH was collapsed with 40 mM of the protonophore benzoate. Mean ± SD; $N = 3$ for Δ*gpmA*, $N = 4$ for WT. **g**, Proton motive force as measured spectrophotometrically with the JC-1 probe in stationary phase *Salmonella* in aerobic MOPS-glucose media. Some cells were pretreated with 400 μM $H_2O_2$ or 20 μM CCCP for 20 min at 37 °C. $N = 3$. **h** Intracellular pH ($pH_i$) of *Salmonella* in macrophages derived from C57BL/6 or $nox2^{-/-}$ mice as measured by fluorescence microscopy. pHluorin$^+$ *Salmonella* were imaged every 2 min p.i for 30 min. Mean ± SEM; $N = 5$ for B6, $N = 7$ for $nox2^{-/-}$. **i**. Phagosomal pH in infected macrophages was monitored for 1 h after infection with FITC- and Alexa-fluor647-labeled *Salmonella*. Mean ± SD $N = 3$. **j** Killing of *Salmonella* by periodate-elicited macrophages from C57BL/6 or $nox2^{-/-}$ mice 18 h after infection. Mean ± SD; $N = 6$. **k** Effects of depolarization on $H_2O_2$ killing. Some specimens were treated with 400 μM $H_2O_2$ and/or 20 μM CCCP. Mean ± SD; $N = 9$ except for CCCP where $N = 3$. Data were analyzed by unpaired, one-tail $t$-test (**a**–**c**, **h**), or one-way (**d**, **k**) or two-way (**e**, **j**) ANOVA. *, **, ***, ****$p < 0.05, 0.01, 0.001, 0.0001$, respectively. Source data are included in Source Data file.

Δ*gpmA Salmonella* to balance $NADH/NAD^+$ and produce ATP. Moreover, Δ*gpmA Salmonella* showed increased *ndh*-encoded NADH dehydrogenase NDH-II enzymatic activity (Fig. 3c). The preferential utilization of NDH-II in Δ*gpmA Salmonella* seems to contribute to its hypersusceptibility to $H_2O_2$ as Δ*gpmA* Δ*ndh Salmonella* were more resistant to $H_2O_2$ than Δ*gpmA* controls (Fig. 3d). Conversely, utilization of the proton-coupled NADH dehydrogenase NDH-I adds to resistance to oxidative stress of aerobic Δ*gpmA Salmonella* as suggested by the hypersusceptibility of a Δ*gpmA* Δ*nuo* double mutant to $H_2O_2$ killing (Fig. 3d). Interestingly, $H_2O_2$ inhibited the proton-coupled NADH dehydrogenase NDH-I (Fig. 3e, Supplementary Fig. 6b). $H_2O_2$ is not likely to directly damage NDH-I[16], although superoxide or peroxynitrite can do so[17,18]. Therefore, the inhibition of NDH-I activity seen in $H_2O_2$-treated *Salmonella* is likely the consequence of indirect inhibition of NDH-I enzymatic activity by ROS derived from $H_2O_2$, an effect that is compounded by the downregulation of *nuo* transcription (Supplementary Fig. 6c). *Salmonella* treated with $H_2O_2$ suffered decreases in ATP content (Supplementary Fig. 6a), possibly reflecting the inhibition of NDH-I.

Both NDH-I and NDH-II oxidize NADH to $NAD^+$, but only NDH-I translocates protons across the cytoplasmic membrane, thereby contributing to ΔpH. Inhibition of the proton-coupled NDH-I isoform during periods of oxidative stress may reduce the intracytoplasmic pH ($pH_i$) of *Salmonella*. To test this possibility, $pH_i$ was measured in *Salmonella* expressing the ratiometric GFP construct pHluorin[19]. Wild-type and Δ*gpmA Salmonella* grown in MOPS-glucose, pH 7.2, maintained a $pH_i$ of 7.8 and 7.7, respectively (Fig. 3f). The addition of 400 μM $H_2O_2$ reduced the $pH_i$ in wild-type *Salmonella*. However, $H_2O_2$ completely dissipated the ΔpH of Δ*gpmA Salmonella* (Fig. 3f). Expression of a *gpmA* allele in trans prevented the massive drop in ΔpH recorded in $H_2O_2$-treated Δ*gpmA Salmonella* (Supplementary Fig. 6d). Comparatively, a Δ*ackA* Δ*pta Salmonella* strain unable to ferment glucose to acetate appears to lack ΔpH (Supplementary Fig. 6e). $H_2O_2$ treatment also collapsed the proton motive force (PMF) in Δ*gpmA Salmonella* to levels seen when wild-type bacteria are treated with the protonophore carbonyl cyanide m-chlorophenyl hydrazine (CCCP) (Fig. 3g). Thus, glycolysis helps preserve ΔpH and PMF during periods of oxidative stress, likely by favoring both the utilization of coupled NDH-I and the

production of fermentation products that help translocate $H^+$ across the membrane.

**NOX2 collapses the ΔpH of intracellular *Salmonella*.** We next tested if NOX2 enzymatic activity affects the ΔpH of intracellular *Salmonella*. Fluorescent microscopy showed that the cytoplasm of intracellular *Salmonella* reaches a $pH_i$ of ~5.6 by 30 min after infection of periodate-elicited peritoneal macrophages from C57BL/6 mice (Fig. 3h). Compared to wild-type bacteria, the cytoplasm of intracellular $\Delta gpmA$ *Salmonella* acidified ~3 times faster. Interestingly, acidification of the cytoplasm of intracellular *Salmonella* was dependent on the enzymatic activity of NOX2 (Fig. 3h). Phagosomes acidified in C57BL/6 and $nox2^{-/-}$ macrophages with similar kinetics, and reached a pH of 5.0 by ~60 min after infection (Fig. 3i). These investigations demonstrate that products of NOX2 dramatically diminish the ΔpH of intracellular *Salmonella*, but glycolytic activity delays the process. The kinetic differences in the dissipation of ΔpH may explain why wild-type *Salmonella* are more resistant than $\Delta gpmA$ controls to the antimicrobial activity of NOX2 (Fig. 3j). It should be noted that collapse of ΔpH does not directly kill *Salmonella*, but it does promote $H_2O_2$ toxicity (Fig. 3k). Together, aerobic *Salmonella* adapt to $H_2O_2$-induced depolarization by maintaining coupled NDH-I enzymatic activity and using glycolysis. This model is supported further by the fact that malate reversed the hypersusceptibility of $\Delta gpmA$ *Salmonella* to $H_2O_2$ (Fig. 1c), while reducing respiratory activity, favoring utilization of NDH-I, and preserving ΔpH and PMF from the toxic effects of $H_2O_2$ (Supplementary Figs. 7a–d).

**Glycolysis protects the Dsb system during oxidative stress.** Next, we examined how reduced electron flow through the membrane boosts resistance to oxidative stress. Thiol-disulfide exchange reactions catalyzed by DsbA and membrane-bound DsbB generate disulfide bonds in periplasmic proteins. Electrons extracted from periplasmic proteins are ultimately delivered by the DsbA/DsbB redox couple into the ETC[20]. We examined whether *Salmonella* undergoing oxidative stress preserves disulfide bond formation in periplasmic proteins by redirecting redox balance to glycolysis. Toward this end, *Salmonella* were transformed with a pBAD plasmid expressing the *E. coli* periplasmic protein PhoA, which requires two disulfide bonds for proper folding and alkaline phosphatase activity. Compared to wild-type *Salmonella*, the enzymatic activity of alkaline phosphatase was markedly inhibited in $\Delta gpmA$ *Salmonella* 30 min after $H_2O_2$ treatment (Fig. 4a). Differences in alkaline phosphatase were neither explained by diminished protein concentrations (Supplementary Fig. 8a) nor by a loss of bacterial viability. Dissipation of the ΔpH by CCCP treatment dramatically diminished the fraction of alkaline phosphatase present in the periplasm (Fig. 4a). As folding of periplasmic proteins by the DsbAB system relies on the ETC, the depolarization induced during 30 min of treatment with CCCP likely explains the massive drop in alkaline phosphatase enzyme activity recorded in these experiments. Together, these findings suggest that the presence of a ΔpH in glycolytic cells maximizes disulfide folding of periplasmic proteins during periods of oxidative stress. The importance of periplasmic disulfide bond formation in resistance of *Salmonella* to oxidative stress is demonstrated by the attenuation of $\Delta dsbB$ *Salmonella* in mice expressing a functional NOX2 (Fig. 4b). The DsbA-DsbB thiol-disulfide exchange system appears to be particularly under stress in $\Delta gpmA$ *Salmonella* with defective glycolytic activity (Fig. 4c, Supplementary Fig. 8b), possible related to the increased respiratory activity of this glycolytic mutant.

We tested whether maintenance of ΔpH across the cytoplasmic membrane impacts DsbA redox status in cells undergoing oxidative stress. To assess the redox state of periplasmic DsbA, thiols were derivatized with the thiol trap 4-acetamido-4′-maleimidylstilbene-2-2′-disulfonic acid (AMS). Western blotting of AMS-treated *Salmonella* expressing a DsbA-3XFLAG-tagged allele revealed the presence of oxidized DsbA proteins in the periplasm of *Salmonella* (Fig. 4d). However, most of DsbA appears to be reduced. The addition of $H_2O_2$ to the cultures did not affect the $DsbA^{red}/DsbA^{oxi}$ ratio in wild-type *Salmonella*; however, the fraction of reduced DsbA dropped in $\Delta gpmA$ *Salmonella* after $H_2O_2$ treatment (Fig. 4d). Growth of $\Delta gpmA$ *Salmonella* on malate prevented unfolding of alkaline phosphatase after $H_2O_2$ treatment, and preserved similar concentrations of oxidized and reduced DsbA redox species (Supplementary Figs. 8c, d). Collapse of ΔpH appears to be responsible for the disappearance of the reduced DsbA fraction in $H_2O_2$-treated $\Delta gpmA$ *Salmonella*, because the addition of the protonophore CCCP lowered the proportion of the reduced state of DsbA in wild-type *Salmonella* (Fig. 4e). Collectively, these data suggest that glycolysis preserves a pool of reduced DsbA that is critical for disulfide bond formation in periplasmic proteins during oxidative stress.

To genetically test the importance of reduced DsbA plays on folding periplasmic proteins, the catalytic site of DsbA was replaced with that of thioredoxin, which when expressed in the periplasm becomes an efficient oxidase[21]. The $DsbA^{TrxA}$ variant was mostly oxidized in the periplasm of *Salmonella* (Fig. 4f), and was able to introduce single disulfide bonds in periplasmic proteins as measured by motility assays (Supplementary Fig. 8e)[22]. However, $DsbA^{TrxA}$-expressing *Salmonella* showed defective folding of periplasmic alkaline phosphatase that requires two disulfide bonds for proper folding. Treatment of $DsbA^{TrxA}$-expressing *Salmonella* with $H_2O_2$ reduced PhoA activity to the low levels recorded in $H_2O_2$-treated $\Delta gpmA$ *Salmonella* (Fig. 4g). Moreover, $DsbA^{TrxA}$-expressing *Salmonella* were not only hypersusceptible to $H_2O_2$ (Fig. 4h), but attenuated in NOX2-expressing mice as well (Fig. 4i). Our investigations indicate that the presence of reduced DsbA is important for protein folding and resistance of *Salmonella* to oxidative stress.

## Discussion

The production of ROS by NOX2 is essential in the innate host response against a variety of bacterial and fungal pathogens. Despite years of intense research, we still have a poor understanding of the mechanism by which NOX2 exerts antimicrobial activity[2,3,5,9,23]. Our investigations show that NOX2 causes a drop in ΔpH in intracellular *Salmonella*. Dissipation of ΔpH together with flow of electrons through the ETC synergize with ROS to kill *Salmonella*. These findings are a major step forward in our understanding of the killing mechanism associated with NOX2. In addition, our investigations unexpectedly show that glycolysis in the bacterium ameliorates the stress oxidants place on the cytoplasmic membrane. Glycolysis shifts redox balance and ATP production from ETC and oxidative phosphorylation to fermentation and substrate-level phosphorylation. Diminished electron flow though the respiratory chain helps folding of periplasmic proteins by the DsbAB thiol-disulfide oxidoreductase system (Fig. 5). This model provides reasonable molecular explanations for why lactate fermentation in the Warburg effect promotes antioxidant defenses of tumor cells[24] and why *Salmonella*, *S. aureus* and *Enterococcus feacalis* with acquired or natural defects in the ETC are remarkably resistant to the respiratory burst[25,26]. However, the glycolytic adaptation in *Salmonella* undergoing oxidative stress is distinct from the Warburg effect,

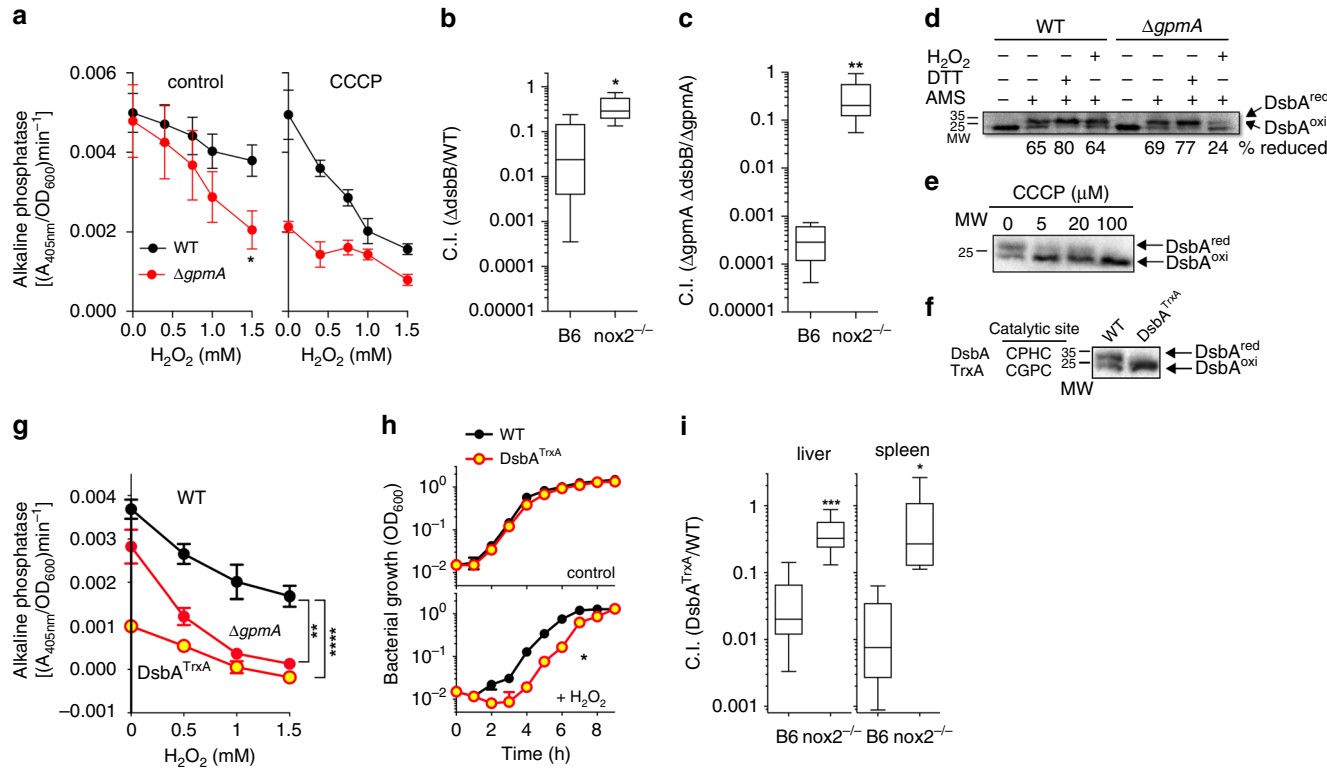

**Fig. 4 Glycolysis facilitates folding of oxidized periplasmic proteins. a** Alkaline phosphatase in *Salmonella* treated for 30 min with either 400 μM $H_2O_2$ or 20 μM CCCP. Mean ± SD; $N = 6$. **b, c** C57BL/6 (B6) and nox2$^{-/-}$ mice were infected with 500 CFU each of the indicated *Salmonella* strains, and the competitive index determined 3 days later. **b** $N = 5$, **c** $N = 8$. **d–f** Western blots of AMS-derivatized DsbA-3xFLAG proteins in WT, Δ*gpmA* or DsbA$^{TrxA}$ *Salmonella* grown aerobically in MOPS-glucose media. Where indicated, the bacteria were treated for 5 min with 10 mM DTT, 1 mM $H_2O_2$ or CCCP. The percentage of reduced DsbA is the mean from three independent experiments as calculated by densitometry. **g** Alkaline phosphatase was measured as described in a. Mean ± SEM; $N = 4$. **h** Aerobic growth of *Salmonella* in MOPS-glucose media in the presence or absence of 100 μM $H_2O_2$. Mean ± SD; $N = 4$. **i** Competitive index of the indicated strains in C57BL/6 and *nox2*$^{-/-}$ mice 3 days after i.p. inoculation. $N = 10$. Data in a and h were analyzed by paired, one-tail *t*-test; data in b, c, i were analyzed by unpaired, one-tail *t*-test. Data in g were analyzed by one-way ANOVA, df = 11, $F = 5.6$. *, **, ***, ****$p < 0.05$, 0.01, 0.001, 0.0001, respectively. All measurements were taken from distinct samples. Whiskers in box plots represent minimal to maxima; 25th and 75th percentiles and median are also represented. Source data are included in Source Data file.

because in addition to favoring glucose fermentation over oxidative phosphorylation it also drives anaplerotic conversion of PEP to oxaloacetate in the TCA. Anaplerotic utilization of PEP might allow synthesis of TCA intermediates for anabolic reactions in *Salmonella* that survive the antimicrobial activity of NOX2[27,28]. Similar anabolic processes may be more widespread in tumor cells than anticipated by the Warburg effect[29,30]. Oxidative TCA also synthesizes NADH to drive lactate fermentation and the proton-pumping activity of NDH-I. The extracellular dissociation of acetate, and possibly the mixed fermentation products lactate and succinate as well, adds to the ΔpH in *Salmonella* undergoing oxidative stress.

An additional effect of diminished electron flow though the respiratory chain upon attack by ROS is to protect folding of periplasmic proteins by the DsbAB thiol-disulfide oxidoreductase system (Fig. 5). The catalytic cysteine residues in DsbA must be kept oxidized by the membrane bound DsbB, whose enzymatic activity relies on the ETC. In *E. coli*, a small fraction of DsbA has consistently been found in its reduced state. Investigators have reasoned that the fraction of reduced DsbA might be a technical artifact. We were surprised to find that over 50% of DsbA is reduced in *Salmonella*. Treatment of *Salmonella* with the uncoupler CCCP completely oxidizes the catalytic cysteine residues of DsbA, a phenotype that is associated with defective protein folding and increased susceptibility to $H_2O_2$. The genetically modified DsbA$^{trxA}$ variant is also overwhelmingly

oxidized. Contrary to what could be predicted from current models, *Salmonella* expressing the predominantly oxidized DsbA$^{trxA}$ variant are not only defective at folding a periplasmic target but hypersusceptible to $H_2O_2$ as well, and show hypersusceptibility to products of NOX2. Together, our investigations indicate that *Salmonella* benefits from retaining a sizable fraction of DsbA in its reduced state. Reduced DsbA may play previously unsuspected functions as a reductase; however, other scenarios cannot yet be ruled out. Overoxidation of DsbA through high electron flow in the ETC may stress other components of the cell envelope thiol-disulfide oxidoreductase system, accounting for misfolding of proteins and predisposing the cell to oxidative stress.

The advent of photosynthesis triggered the accumulation of $O_2$ on earth. Tetravalent reduction of molecular oxygen ($O_2$) by cytochromes of the electron transport chain (ETC) provided a surplus of energy that powered the explosion of animal forms in the Ediacaran period ~650 million years ago[31–33]. However, the univalent reduction of $O_2$ yields superoxide, a precursor of downstream ROS whose toxicity is exploited by NOX2 in the host response. The energetic advantages associated with complete reduction of $O_2$ by terminal cytochromes contrast with the toxicity of ROS generated by the partial reduction of this diatomic gas. Paradoxically, glycolysis present in bacteria before $O_2$ was used by respiratory cytochromes offers a solution to the toxicity of congeners of this paramagnetic species. The glycolytic/ETC

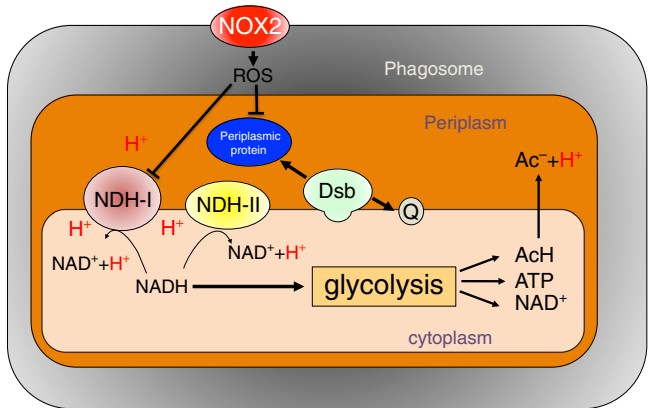

**Fig. 5 Model for the metabolic adaptations that ameliorate NOX2 killing of intracellular *Salmonella*.** Reactive oxygen species (ROS) produced by NOX2 in the phagosome target the proton-coupled NDH-I isoform of NADH dehydrogenase of *Salmonella*. Oxidative stress also diminishes transcription of *nuo* genes encoding NDH-I. Together, the negative effects oxidative stress has on the expression of NDH-I contributes to the dissipation of ΔpH across the bacterial membrane. Collapse of ΔpH negatively impacts the function of the oxidoreductase Dsb system, which introduces disulfide bonds among in cysteine residues of periplasmic proteins (blue). Collapse of ΔpH, defective Dsb function and oxidation of cell envelope proteins catastrophically synergize to mediate NOX2-dependent killing of intracellular *Salmonella*. In response to the selective pressure oxidative stress imposes on the ETC, *Salmonella* shuttles of NADH reducing power to fermentation, thereby not only diminishing electron flow through ETC but also allowing for proper folding of periplasmic proteins by Dsb oxidoreductases that delivered the extracted reducing power into the quinones (Q) of the ETC. Fermentation of glucose also provides ATP synthesis by substrate-level phosphorylation, balances NAD⁺/NADH, and helps maintain ΔpH across membrane via dissociation of products such as acetic acid (AcH).

adaptation described herein must be added to superoxide dismutases, catalases, peroxidases, thioredoxins, glutaredoxins, glutathione, and endonucleases as a key component that protects bacteria against oxidative stress[4,6].

## Methods

**Bacterial strains, media and growth conditions.** Bacterial strains, plasmids and primers used are listed in Supplementary Tables 1 and 2. *Salmonella enterica* serovar Typhimurium ATCC 14028 was used as the wild-type strain in all experiments and as a background for all targeted gene deletions. *Salmonella* mutants were constructed by the λ Red-recombinase gene replacement method[34], and mutations were moved by P22-mediated phage transduction. Δ*gpmA Salmonella* was complemented by integrating the wild-type *gpmA* ORF along with its promoter at the put locus. Nucleotide substitutions of *gpmA* were generated with the QuikChange site-directed mutagenesis kit (Stratagene, La Jolla, CA). Strains were routinely grown at 37 °C in LB broth[35]. MOPS (morpholino propanesulfonic acid) minimal medium[36], pH 7.2, supplemented with 0.4% D-glucose was used; however, where indicated, glucose was replaced with 0.4% glycerol, 0.48% sodium pyruvate, 0.45% sodium succinate, 0.54% sodium acetate, 0.44% malic acid, 0.54% sodium fumarate. For anaerobic assays, cultures were grown overnight in an anaerobic chamber in LB broth followed by culture in MOPS minimal medium for 2 h prior to H₂O₂ challenge.

**Tn-seq screen.** Construction of the barcoded Tn5-seq library in *S. enterica* sv Typhimurium strain 14028 s has been described elsewhere[37,38]. The specimens were grown for 20 h in LB broth at 37 °C with vigorous shaking. An aliquot of the culture was stocked in 10% glycerol at −80 °C as the input, and another aliquot was diluted 1:100 into 10 mL of MOPS minimal media supplemented with glucose and/ or Casamino acids. The cultures were grown for 2.5–4 h to an approximate OD₆₀₀ of 0.2. Cultures were split into 2 mL aliquots that were either left untreated or were challenged with 2 mM H₂O₂. After 2 h of challenge, the cells were transferred to LB broth and grown for 20 h before storing in 10% glycerol at −80 °C. The procedure was performed in biological replicates on different days. Samples that had undergone the same treatment were very similar. Sequencing and statistical

analysis (https://bioconductor.org/packages/release/bioc/html/DESeq2.html) was done as described[11,37,39].

**H₂O₂ survival assays.** H₂O₂ killing assays were carried out with *Salmonella* grown overnight in LB broth and then subcultured and grown aerobically to log phase in MOPS minimal media supplemented with glucose or the indicated carbon sources. Briefly, bacterial cultures grown to OD₆₀₀ of 0.4 in MOPS minimal media were diluted to 10⁶ CFU in 96-well plates and challenged with 400 μM H₂O₂ for 2 h at 37 °C. The number of bacteria that survived challenge was enumerated by plating serial dilutions. Percent survival was calculated as [CFU from H₂O₂-treated sample/CFU from untreated sample]. Where indicated, *Salmonella* were treated at the time of challenge with 20 μM CCCP, 1 mM deferoxamine or dipyridyl. Anaerobic H₂O₂ killing assays were performed with cells resuspended in MOPS-glucose media for 2 h from stationary cultures grown for 24 h in LB broth in an anaerobic chamber. Some of the anaerobic cultures were grown in LB broth containing 50 mM of the terminal electron acceptor sodium nitrate. To measure the effects of oxidative stress on growth, stationary phase bacteria grown in LB broth were diluted to ~10⁴ CFU in MOPS media and challenged with 50 μM H₂O₂. Growth of untreated and H₂O₂-treated *Salmonella* was recorded on a Bioscreen C plate reader (Growthcurves USA, Piscataway, NJ) for 30 h with continuous shaking at 37 °C.

**Metabolomics.** Five biological replicates collected on two different days were grown aerobically in MOPS-glucose media to OD₆₀₀ of 0.6. Some specimens were treated with 2.5 mM H₂O₂ for 30 min. Under these conditions neither wild-type nor Δ*gpmA Salmonella* experienced loss of viability; however, viability dropped 7-fold by 2 h of H₂O₂ treatment. The metabolites were analyzed by Metabolome Inc., Durham, NC, using ultrahigh performance liquid chromatography/tandem mass spectrometry (UPLC-MS/MS) optimized for basic, acidic and hydrophobic metabolites. The data were normalized to internal dsDNA concentrations, and differences within strains and treatments were identified by two-way ANOVA ($p < 0.05$) and a false discovery rate of <0.1.

Two mL bacterial cells from overnight aerobic LB cultures were washed twice in PBS and resuspended in 125-mL flasks containing 40 mL MOPS-glucose medium at 37 °C. Cultures were grown aerobically for 1 h with shaking followed by 30 min treatment with 2.5 mM H₂O₂. Untreated specimens were collected for comparison. Cells chilled on ice were harvested by centrifugation. Bacterial pellets washed in PBS were collected and stored at −80 °C. To recover chemically diverse metabolites while removing proteins, the specimens were precipitated with methanol by vigorous shaking for 2 min in a Glen Mills Genogrinder 2000. The samples were centrifuged and the resulting supernatants (MicroLab STAR® system) were split into fractions for analyses on: A) two separate reverse-phase (RP)/UPLC-MS/MS with positive ion mode electrospray ionization (ESI), B) RP/UPLC-MS/MS with negative ion mode ESI, and C) HILIC/UPLC-MS/MS with negative ion mode ESI platforms by Metabolon, Inc (NC, USA). Samples were placed briefly on a TurboVap (Zymark) to remove the organic solvent. The sample extracts were stored overnight under nitrogen before preparation for analysis.

Bacterial extract was dried and reconstituted in solvents compatible to each of the different analyses platforms. Each reconstitution solvent contained a series of internal standards at fixed concentrations to ensure injection and chromatographic consistency. For UPLC-MS/MS, samples were separated using a Waters Acquity UPLC (Waters, Millford, MA) instrument with separate acid/base-dedicated 2.1 mm × 100 mm Waters BEH C18 1.7 μm particle columns heated to 40 °C and analyzed using a Q-Exactive mass spectrometer (Thermo Fisher Scientific, Inc., Waltham, MA), which consisted of an electrospray ionization (ESI) source and Orbitrap mass analyzer. One aliquot, optimized for acidic positive ion conditions, was gradient eluted using water and methanol containing 0.05% perfluoropentanoic acid (PFPA) and 0.1% formic acid, whereas hydrophobic compounds in another aliquot were eluted using methanol, acetonitrile, water, 0.05% perfluoropentanoic acid (PFPA) and 0.01% formic acid. A third aliquot was analyzed for basic extracts using a separate dedicated C18 column and gradient eluted using methanol and water with 6.5 mM ammonium bicarbonate, pH 8.0. The fourth aliquot was analyzed via negative ionization following elution from a HILIC column (Waters UPLC BEH Amide 2.1 × 150 mm, 1.7 μm) using a gradient consisting of water and acetonitrile with 10 mM ammonium formate, pH 10.8. The MS instrument scanned 70–1000 m/z and alternated between MS and MS2 scans using dynamic exclusion with approximately 6 scans per second. Metabolites were identified by automated comparison of the ion features in the experimental samples to a reference library of chemical standard entries that included retention time, molecular weight (m/z), preferred adducts, and in-source fragments as well as associated MS spectra.

Following normalization to dsDNA concentration and log transformation, mixed model ANOVA contrasts were used to identify metabolites that differed significantly ($p \leq 0.05$) or approached significance ($0.05 < p < 0.10$) between experimental groups. Analysis by two-way ANOVA identified biochemicals exhibiting significant interaction and main effects for experimental parameters of strain (wild-type vs. Δ*gpmA*) and treatment (control vs. H₂O₂). Multiple comparisons were accounted for by estimating the false discovery rate using q-values.

**Polarographic $O_2$ and $H_2O_2$ measurements.** Consumption of $O_2$ was measured using an ISO-OXY-2 $O_2$ sensor attached to an APOLLO 4000 free radical analyzer (World Precision Instruments, Inc., Sarasota, FL) as described in ref. [25]. Briefly, one mL of *Salmonella* grown aerobically to $OD_{600}$ O.4 in either MOPS-glucose or -malate were rapidly withdrawn, vortexed for one minute and immediately recorded for $O_2$ consumption. A two-point calibration for 0 and 21% $O_2$ was done as per manufacturer's instructions. $H_2O_2$ consumption was measured similarly using a $H_2O_2$ analyzer probe, after treating 1 mL culture with 100 μM $H_2O_2$ for a minute. ΔkatEGN was used as a negative control.

**Enzyme assays.** NADH dehydrogenase I (*nuo*) and NADH dehydrogenase II (*ndh*) enzyme assays were performed as described[40,41] with slight modifications. Briefly, 50 mL of bacterial cells grown aerobically in either MOPS-glucose or -malate to $OD_{600}$ of 0.5 were harvested and washed with ice-cold 50 mM MES buffer containing 10% glycerol, pH 6.0. Bacterial pellets were resuspended in 10 mL of the same buffer and lysed by sonication. Cell debris was removed by centrifugation at $17,000 \times g$ for 20 min at 4 °C. Supernatants were split in 5 mL aliquots in two ultra-centrifuge tubes and spun at $100,000 \times g$ for 2 h at 4 °C to harvest inverted membrane vesicles. Pellets were resuspended in either 1 mL cold 50 mM MES buffer with 10% glycerol, pH 6.0, for NDH-I or 50 mM $KPO_4$ buffer with 5 mM $MgSO_4$, pH 7.6, for NDH-II assays, respectively. Membrane vesicles assigned for NDH-II activity were stored on ice overnight to deactivate NDH-I. NDH-I activity was measured kinetically at $A_{340}$ with 200 μM of deamino-NADH, whereas NDH-II activity was monitored similarly after incubation on ice in the presence of 3 mM KCN, 0.1 mM plumbagin and 200 μM β-NADH. Total protein was estimated using Pierce BCA protein reagent and specific activity was determined per mg of protein. For determining NADH dehydrogenase activity post $H_2O_2$ challenge, *Salmonella* grown aerobically in MOPS-glucose were challenged with 2 mM $H_2O_2$ for 2 h under shaking, followed by isolation of membrane vesicles as described above. To determine direct effects of $H_2O_2$ on NADH dehydrogenase activity in vitro, vesicles isolated from wild-type *Salmonella* grown for 20 h in LB broth were exposed to 100 or 200 μM $H_2O_2$. Membrane fractions resuspended in either MES or $KPO_4$ buffer were incubated for 10 min at 30 °C with $H_2O_2$. Substrates, d-NADH or β-NADH, were then added and absorbance recorded at 340 nm for 30 min.

The enzymatic activity of ectopically expressed alkaline phosphatase (PhoA) was determined by measuring the rate of para-nitrophenyl phosphate (pNPP) hydrolysis at 410 nm as described in ref. [42]. Overnight aerobic cultures expressing *E. coli* PhoA, supplemented with 0.2% L-arabinose, were sub-cultured in fresh MOPS-glucose or -malate media and grown aerobically to $OD_{600}$ of 0.5. Where indicated some of the specimens were treated for 30 min with $H_2O_2$ and/or CCCP. Then, 100 μL of cells were mixed with 100 μL of 1 M Tris, pH 8.0, 1 mM zinc chloride and 0.01% SDS buffer in 96-well microtiter plates. Reactions were initiated by addition of 25 mM pNPP and rate of hydrolysis measured for 1 h at 37 °C. PhoA enzymatic activity corresponds to the initial linear slope normalized to bacterial density as measured at $OD_{600}$.

**NADH(P)/NAD(P)$^+$ quantification.** Estimations of intracellular pyridine nucleotide pools were carried out according as described[25] with slight modifications. Briefly, NADH and NAD$^+$ were extracted in 0.2 M NaOH and 0.2 M HCl, respectively, from 1 mL of bacteria grown aerobically in either MOPS-glucose or –malate media grown to $OD_{600}$ of 0.5. The reaction mixtures contained 1 M bicine, pH 8.0, 40 mM EDTA, 16 mM phenazine methosulfate and 4.2 mM 3-(4,5-dimethylthiazol-2-yl)−2,5-diphenyltetrazolium bromide. For NAD$^+$/NADH, 100% ethanol and 1 mg/ml alcohol dehydrogenase were added to the reactions, which were read at $A_{570}$. The concentration of NADH and NAD$^+$ was measured by the thiazolyl tetrazolium blue cycling assay and calculated by regression analysis of known standards. Nucleotide concentrations were corrected for bacterial density as estimated spectrophotometrically at $OD_{600}$. Intracellular NADH and NAD$^+$ concentrations were calculated based on a bacterial cell volume of $10^{-15}$ liters.

**Intracellular pH determination.** To estimate intracytoplasmic pH of *Salmonella*, the pH sensitive pHluorin driven from the $P_{BAD}$ promoter was used[19]. Overnight bacterial cultures aerobically grown in the presence of 0.2% L-arabinose were resuspended to $OD_{600}$ of 0.4 in 3 mL of MOPS-glucose or -malate media, pH 7.2. After an equilibration of cells in the MOPS media for 30 min, the cultures were scored for fluorescence using a Shimadzu R5300C spectrofluorometer with an excitation wavelength scan from 300 to 490 nm and emission at 510 nm for 5 min. The cells were then treated with 400 μM $H_2O_2$ and the excitation spectra measured for an additional 25 min. At the end of the time course, 40 mM of the protonophore potassium benzoate were added to equilibrated pH$i$ (internal) and pH$e$ (external). To determine the value of internal pH from the obtained fluorescence ratio (405/488 nm), pHluorin-expressing *Salmonella* were incubated in MOPS media pH 5.0 to 9.0 in the presence of protonophore potassium benzoate. A Boltzmann sigmoid best-fit curve was applied to the plot of fluorescence ratio and the known pH values obtained using the standards. The two excitation $\lambda_{405}$ and $\lambda_{488}$ peaks of pHluorin are directly proportional to changes in pH.

We also measured the pH$_i$ of intracellular *Salmonella*. Briefly, sodium periodate-elicited peritoneal macrophages[43] from C57BL/6 and nox2$^{-/-}$ mice

seeded on a glass bottom Cellview cell culture dish (Greiner bio-one) were infected at an MOI of 50:1 with pHluorin-expressing *Salmonella* grown aerobically overnight in LB broth with 0.2% L-arabinose. After 20 min of infection, the infected monolayers were imaged in a Olympus fv1000 confocal laser scanning microscope with consecutive excitations at 405 and 488 nm and emission at 525/440 nm. Multiple Z sections were taken, and the maximum projections were analyzed using ImageJ to determine the mean fluorescence intensity of intracellular bacteria at 405 and 488 nm. To convert the 405/488 ratio values to cytoplasmic pH, a standard curve was generated by recording the GFP fluorescence of intramacrophagic bacteria after equilibrating the intra-bacterial pH with MOPS pH 5–9 containing 40 mM of the protonophore potassium benzoate. The ratio values were fitted with a Boltzmann sigmoid curve and solved for actual pH values as described above. All images were corrected for background using adjacent cell-free areas.

**Determination of phagosomal pH.** The phagosomal pH of *Salmonella*-containing vacuoles was determined as described[44,45]. *Salmonella* were labeled with 1 and 0.1 mg/ml of the pH sensitive and insensitive FITC and Alexa-fluor-647 probes, respectively. Periodate-elicited macrophages from C57BL/6 or nox2$^{-/-}$ mice were infected at an MOI of 50 with dually labeled *Salmonella*. After 20 min of infection, the monolayers were washed twice with DMEM media containing 10% FBS and harvested over time for flow cytometric analysis. Phagosomal pH was monitored every 10 min for 1 h after infection by recording mean fluorescence intensity at 488 and 633 nm on a Cytoflex S (Beckman Coulter) flow cytometer. As control, the ΔpH across phagosomal membranes was collapsed after 20 min treatment of macrophages with 50 nM bafilomycin.

**Determination of proton potential.** For assessing membrane potential, 0.3 O.D of exponentially growing aerobic *Salmonella* were adjusted to ~$10^8$ CFU/mL. Bacterial cell membrane potential was estimated using the J-aggregate forming lipophilic cation 5,5′,6,6′-tetrachloro-1,1′,3,3′-tetraethylbenzimidazocarbocyanine iodide (JC-1) dye (Molecular Probes, Eugene, OR)[46]. Fluorescence intensity at 530 and 595 nm was measured with an excitation at 488 nm by a Shimadzu R5300C spectrofluorometer. Briefly, $10^8$ CFU/mL *Salmonella* in PBS either untreated or treated with 400 μM $H_2O_2$ were exposed to 0.5 μL of 5 mg/mL JC-1) and incubated for 20 min at 37 °C. Where indicated, 20 μM of the protonophore CCCP were added to the untreated or $H_2O_2$-treated *Salmonella*. Scanning emission spectra (500–650 nm) after excitation at 488 nm was recorded 20 min after the bacteria were incubated with the probe.

**Mouse infections.** All animal experiments were performed with ethical approval from the University of Colorado School of Medicine Institutional Animal Care and Use Committee. Five-mice per cage were housed in static cages under BSL2 conditions. Males and females, 6-8 week-old C57BL/6 or congenic nox2-deficient mice were inoculated intraperitoneally with ~200 CFU of stationary phase *Salmonella* grown for 20 h in LB broth. Mouse survival was monitored for 2 week. Mice manifesting signs of distress (i.e., low spontaneous activity and ruffled coat) were humanely euthanized by $CO_2$ inhalation followed by cervical dislocation. For competition assays, mice were inoculated i.p. with 500 CFU each of strain 1 and strain 2. Spleens and livers were harvested 3 days after inoculation and CFU quantified by serial dilution on plates containing the appropriate antibiotics. Competitive index is defined as the ratio of mutant and wild-type strains within the output divided by the ratio of mutant and wild-type strains within the input.

**AMS assay.** 4-acetamido-4-maleimidylstilbene-2,2′-disulfonic acid (AMS) trapping[47] and Western blotting were used to determine the redox state of DsbA. *Salmonella* expressing the *dsbA*::3XFLAG allele were cultured aerobically to stationary phase in LB broth at 37 °C. Overnight cultures washed and diluted in MOPS-glucose or -malate media to $OD_{600}$ of 0.5 and were incubated aerobically at 37 °C for 15 min with shaking. Where indicated, specimens were treated with 10 mM DTT or 1 mM $H_2O_2$. After 5 min, samples were treated for 30 min with 15% trichloroacetic acid and proteins collected by centrifugation. The resulting pellets were air-dried and dissolved in 40 μl of freshly made 1 M Tris-HCl, pH 8.0, 0.1% SDS, 1 mM EDTA, and 15 mM AMS (Molecular Probes). Controls were not treated with AMS. Samples were incubated at 37 °C for 1 h in the dark followed by addition of 5x SDS loading dye without reducing agent. Proteins were separated by 12% SDS-polyacrylamide gel electrophoresis. Reduced and oxidized forms of DsbA were visualized on Western blots using mouse anti-FLAG antibody (Catalog number F1804, Sigma-Aldrich, St Louis, MO). AMS reacts specifically and irreversibly with cysteine sulfhydryl groups, resulting in a 0.5 kDa mass change per reduced cysteine residue.

**Statistical analysis.** Statistical analyses were performed using GraphPad Prism 5.0b Software. One-way and two-way ANOVA, and t-tests were used. Results were determined to be significant when $p < 0.05$.

**Reporting summary.** Further information on research design is available in the Nature Research Reporting Summary linked to this article.

## Data availability

The source data underlying Figs. 1a, 2a, b, and Supplementary Fig. 3 are provided as Source Data files at https://figshare.com/s/c0144e4f2f617454ffc7, https://figshare.com/s/d361f10bfd4bab217839, https://figshare.com/s/66439f6e180011832f7b https://figshare.com/s/66ad21194d76adb17d8e. All other source data can be found in the Source Data excel file. Other datasets generated during the current study are available from the corresponding author on request.

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

## Acknowledgements

This research was funded by VA Merit Grant BX0002073, and NIH grants AI054959, AI136520, AI118223, and AI052066. SP, PD and MM were supported, in part, by USDA grants 2017-67017-26180 and 2017-67015-26085, NIAID contract number HHSN272200900040C, and NIH grant R01AI136520. We thank Dr. Jessica Jones-Carson for providing the mice used in this study and for discussions during the preparation of this paper. All unique materials are readily available to any investigators for noncommercial use.

## Author contributions

S.C., L.F., and A.V-T. designed the study. L.L., S.C., and S.P. constructed strains, plasmids, and Tn5 libraries. S.C., L.F., L.L., J-S.K., S.P., and A.V-T. performed experiments. S.C., L.F., S.P., P.D., M.M., and A.V-T. analyzed the data. A.V-T. wrote the paper with help from the other authors.

## Competing interests

The authors declare no competing interests.
