## [Peer Review File · Nature Communications]

Reviewers' comments:

Reviewer #1 (Remarks to the Author):

Chakraborty et al show that a mutant lacking the metal-independent glycerophosphomutase GpmA is sensitive to hydrogen peroxide. The data presented are intriguing, but you never really present a coherent model to explain the results. This lack of any mechanistic explanation means that this is a series of unexplained phenomena.

1. The title makes little sense, even after reading the paper.
2. Line 82 and elsewhere. Granted, the gpmA mutant is sensitive to H₂O₂, but to then conclude that glycolysis has "antioxidant functions" implies mechanisms to directly combat ROS. Rephrase.
3. Line 82. "The key..." This sentence is confusing here. First, your point is really not clear, and it changes the focus from your line of inquiry.
4. The results in Fig 1 imply that flux through the respiratory chain is required for the phenomenon, but you never provide an explanation to the reader.
5. As for the point above, there really doesn't seem to be a significant difference between the wildtype and gpmA mutant in nitrate, so the entire conclusion is suspect.
6. Fig 2. The gpmB mutation does not confer a phenotype, but the gpmA mutation does, implying that the flux through the pathway is via GpmA. Yet, there is no effect on downstream glycolytic intermediates in the GpmA mutant?
7. What is your explanation of why the gpmB mutation does not confer a phenotype?
8. Fig 2 legend. You state that red arrows indicate "defective anaplerotic utilization of PEP". What does "defective" mean? Are these compounds increased or decreased?
9. Line 106. You state that "Salmonella...undergo a glycolytic switch to balance NADH/NAD and generate ATP." It is not clear what you really mean.
10. Line 129 – the "payoff phase" is jargon. Rephrase.
11. Line 135. You are surprised that the gpmA mutant has low HADH and higher ATP than wildtype, given that it is defective in lactate production. But these measurements were taken in the absence of H₂O₂, whereas the lactate decrease was seen in treated cells. Not clear that you can directly compare them.
12. Line 162. What is your model to explain how GpmA would influence your proposed dissipation of DpH?
13. Fig S5. One of your primary conclusions is that H₂O₂ inactivates NDH1. The rate of this in vitro reaction suggests that you are simply oxidizing any thiols in the protein ([https://doi.org/10.1016/S0891-5849\(99\)00051-9](https://doi.org/10.1016/S0891-5849(99)00051-9)). More importantly, Imlay and colleagues have shown that NDH1 is NOT damaged in vivo by H₂O₂ even at 5 mM (doi: 10.1074/jbc.M607646200).
14. Your interpretation of the DsbA data seems backwards. Most importantly, the previous evidence shows that the role of DsbA is to introduce disulfide bonds into periplasmic proteins. DsbC and G act to repair oxidized thiols.

Reviewer #2 (Remarks to the Author):

In this manuscript, the authors set out to answer the long-debated question of how NADPH oxidase kills microbes. The authors present sound data supporting the hypothesis that NADPH oxidase collapses the ΔpH of intracellular *Salmonella* and synergizes with hydrogen peroxide to maximize antimicrobial activity. Additionally, the authors show that a microbial adaptation to resist the respiratory burst lies in the shift from ETC to glycolysis.

The experiments are presented logically and support the author's hypothesis in a -given the complex nature of the subject- relatively easy to understand way. The experiments were performed rigorously and to high scientific standard. The authors employ mutants in glycolysis enzymes in their study and pleiotropic effects could have easily masked relevant findings. However, the authors present numerous highly relevant controls for their experiments. The findings of the study are novel and should interest a broad readership.

Specific comments:

Statistical analyses should be provided for: mouse experiments, FigS3 (according to M&M analysis was done by ANOVA, but is not indicated in the figure), 3d (no statistical difference to *gpmA* *ndh*?).

M&M describes complementation of mutant strains. For most experiments, multiple mutants were analyzed that showed similar effects, making off-target effects unlikely. However, key experiments with complemented strains should be presented.

If in 4d and S7d representative blots are shown, then more should be available to show the average of densitometric analyses.

Admittedly, I could not follow rationale or conclusions for Fig. 4g-i. A few additional sentences might clarify this.

Y axis are frequently labeled in a rather unconventional way that requires conversion in order to interpret. Readers will be familiar with $\text{OD}_{600}=1$; presenting it as 10^0 is unnecessarily confusing. I do not see any reason for presenting survival logarithmically. And again, presenting it as 10^{-1} to 10^2 is confusing to the reader.

Fig. 1c: The reduced survival with succinate should be explained.

Fig 2A could additionally serve as helpful reminder as to where in the pathway the enzymes relevant for Fig. 1 and other figures are active. Please add the few enzymes that are relevant for the paper. This should be possible without being too distracting for the findings of Fig. 2A and could facilitate the reader's understanding.

Could the authors provide the reasoning behind different concentrations of H_2O_2 used? 400 μM for most assays, 2.5mM for metabolomics. After 2h with 400 μM only 0.2% of *gpmA* is alive, after 30min with 6x this concentration I would assume that cell viability is affected, yet M&M states that no viability loss was seen?

Fig. S3.: Check descriptions in text: succinate does not seem to be lower in untreated *gpmA* compared to WT. Also 3-phosphoglycerate (as expected) higher in *gpmA*.

Check color and legend for Fig. 2d. The red should be purple/blue as well?

3f: please describe the effect of addition of benzoate. It is mentioned in the M&M, but as appears in a main figure, it should be explained.

4a: why is gpmA alkaline phosphatase at 0mM H₂O₂ so much lower with CCCP than WT?

Reviewer #3 (Remarks to the Author):

This comprehensive study of *Salmonella typhimurium* by the authors provides compelling evidence for the gene *gpmA*, which encodes for glycerophosphomutase, having a key antioxidant role by facilitating a switch to glycolysis during oxidative stress. The finding is demonstrated to be relevant to *Salmonella* pathogenesis using a mouse infection model. It is generally understood that glycolysis benefits cells under oxidative stress, particularly in cancer. The novel finding of this study is identifying *gpmA* as a key player in the metabolic switch and the discovery of its role in oxidative stress protection and pathogenesis. Overall, this is a very thoughtful and significant contribution that is of high importance in the field of metabolism and cellular stress response. The conclusions are supported by convincing and high quality data and the manuscript significantly advances knowledge of metabolic reprogramming during stress.

Some suggestions/corrections.

1. Line 494, Figure 1C. What *Salmonella* strain is being used? Please confirm it is the *gpmA* mutant. This also needs to be made clear in Figure 3e.
2. Line 66, deferroxamine did not protect the hydrogen peroxide killing of the *gpmA* mutant. Were other antioxidants tested as well? e.g, N-acetylcysteine
3. In Figure 2, define *ldhA* (D-lactate dehydrogenase gene)
4. What are the red squares in Figure 2d? *gpmA* mutant strain?
5. Figure 4d, Need to indicate whether hydrogen peroxide is present or not in lane 3 of the figure.
6. Figure 4J-The reaction arrow for *DsbB* is in the wrong direction relative to *DsbA*.
7. Consider using the enzyme nomenclature of phosphoglycerate mutase instead of glycerophosphomutase.
8. Add a comment about whether there are significant changes in the GSH/GSSG ratio.
9. Provide more of an explanation for the higher ATP levels found in the *gpmA* mutant cells relative to WT (line 136 and Fig. S5a). Also, what is the impact of hydrogen peroxide treatment on the ATP levels in the *gpmA* mutant cells relative to the WT strain?

Reviewer #1 (Remarks to the Author):

Chakraborty et al show that a mutant lacking the metal-independent glycerophosphomutase GpmA is sensitive to hydrogen peroxide. The data presented are intriguing, but you never really present a coherent model to explain the results. This lack of any mechanistic explanation means that this is a series of unexplained phenomena.

Response: NOX2 is essential in the innate host response against a variety of bacterial and fungal pathogens. Despite years of intense research, we still have a poor understanding of the mechanism by which NOX2 exerts antimicrobial activity. The goal of these investigations was to find out how NOX2 kills organisms using Salmonella as a model. Our investigations prove that NOX2 causes a drop in ΔpH in intracellular Salmonella. Dissipation of ΔpH and enhanced flow of electrons synergizes with ROS to mediate killing. These findings are a step forward in our understanding of killing by the highly conserved NOX2. Given the tremendous selective pressure imposed by NOX2, the bacterium attempts to counter this attack. Our investigations prove that glycolysis helps Salmonella adapt to the depolarization of the cytoplasmic membrane. As a result, there is an inevitable shifting of redox balance and ATP production from ETC and oxidative phosphorylation to fermentation and substrate level phosphorylation, thereby diminishing flow of electrons through the respiratory chain. By identifying an additional way that ROS affects the bacterium, and a major way that bacterial metabolism is altered, these studies represent an advance in the understanding of the metabolic adaptation that increases fitness of intracellular bacteria during the intense oxidative stress associated with NOX2. We have modified the title and abstract to more clearly convey the importance of our investigations. A more comprehensive presentation of our novel findings can be found in legend of Fig. 5 and Discussion.

The title makes little sense, even after reading the paper.

The title has been changed to more clearly reflect the novel findings of our investigations. It now reads "Glycolytic reprogramming in Salmonella counters NOX2-mediated dissipation of ΔpH ."

Line 82 and elsewhere. Granted, the gpmA mutant is sensitive to H_2O_2 , but to then conclude that glycolysis has "antioxidant functions" implies mechanisms to directly combat ROS. Rephrase.

The reviewer is correct that glycolysis is not a direct antioxidant. As suggested, the text has been modified to clearly convey the idea that glycolysis is needed for optimal resistance to oxidative stress.

Line 82. "The key..." This sentence is confusing here. First, your point is really not clear, and it changes the focus from your line of inquiry.

As requested, the sentence has been edited.

The results in Fig 1 imply that flux through the respiratory chain is required for the phenomenon, but you never provide an explanation to the reader.

Experiments in Fig. 1 establish that flow of electrons through the ETC is important for the antimicrobial activity of H_2O_2 against Salmonella. These findings considerably advance our understanding of the mechanism by which this biologically relevant ROS imparts cytotoxicity. As the reviewer comments, at this point of the paper we still don't know why flow of electrons through the ETC enhances killing by H_2O_2 , which is not presented until Figs. 3 and 4. The model summarizing our findings is presented in revised Fig. 5 and Discussion.

As for the point above, there really doesn't seem to be a significant difference between the wildtype and *gpmA* mutant in nitrate, so the entire conclusion is suspect. In the original graph, the *gpmA* mutant was trending to be more susceptible to H₂O₂ than wild-type *Salmonella* under anaerobic growth when the terminal electron acceptor nitrate was included in the media. We have repeated the assay (n=27) and show that the trend is in fact statistically significant (p < 0.01) (revised Fig. 1f). It should be noted that anaerobic wild-type *Salmonella* are also significantly (p < 0.0001) more susceptible to H₂O₂ killing when grown with nitrate. This means that even in wild-type bacteria increased flow of electrons through the ETC results in enhanced H₂O₂ cytotoxicity. We have tested if electron flow is responsible for the increased susceptibility of wild-type *Salmonella* under these conditions, and have found that mutations in either *nuo* or *ndh* completely protect against H₂O₂ killing. These data, which are shown in Fig. S1f, are consistent with the idea that elevated electron flow through the ETC predisposes *Salmonella* to H₂O₂ toxicity. The Results have been edited according to the new findings.

Fig 2. The *gpmB* mutation does not confer a phenotype, but the *gpmA* mutation does, implying that the flux through the pathway is via GpmA. Yet, there is no effect on downstream glycolytic intermediates in the *GpmA* mutant?

Glycolysis is peppered with essential genes that are not amenable to mutagenesis in *Salmonella*. Despite its richness in essential genes, a few loci can be mutagenized. Remarkably, we found that transposons in *gpmA* or pyruvate kinase *pykF* (Table S2) sensitized *Salmonella* to enhanced H₂O₂ killing. Using a targeted approach, we further found that a double *pfkAB* mutant deficient in phosphofructokinase is also hypersusceptible to products of NOX2 (Fig. 1d). Together, our investigations demonstrate the importance of glycolysis in resistance of *Salmonella* to oxidative stress. It is possible that the reasons why it took so long to uncover this crucial phenotype are related to the fact that so many genes are effectively essential in the affected pathway and to the fact that there was no a priori reason to test whether this pathway participates in resistance to oxidative stress.

What is your explanation of why the *gpmB* mutation does not confer a phenotype? *GpmB* is a manganese-dependent isoform of phosphoglycerate mutase. Metabolism of manganese is under selection in cells undergoing oxidative stress, as illustrated by the disadvantage of *mntH* Tn mutants deficient in the high affinity manganese uptake system in a screen with H₂O₂ (Fig. 1a). We show that *Salmonella* needs the manganese-independent *GpmA* phosphoglycerate mutase isoform under oxidative stress, a condition that seems to limit manganese availability in the cell. This information has been added to the revised manuscript.

Fig 2 legend. You state that red arrows indicate "defective anaplerotic utilization of PEP". What does "defective" mean? Are these compounds increased or decreased? Malate and fumarate are in lower concentrations in Δ *gpmA* *Salmonella* compared to wild-type controls, whereas PEP is increased in the former. The text has been edited accordingly.

Line 106. You state that "*Salmonella*...undergo a glycolytic switch to balance NADH/NAD and generate ATP." It is not clear what you really mean. Sorry for the confusion. We mean that *Salmonella* undergoing oxidative stress are better off when glycolysis and fermentation are functional. The text has been modified to more clearly communicate our thoughts.

Line 129 – the “payoff phase” is jargon. Rephrase.

As requested, we have erased payoff phase from the sentence.

Line 135. You are surprised that the *gpmA* mutant has low HADH and higher ATP than wildtype, given that it is defective in lactate production. But these measurements were taken in the absence of H₂O₂, whereas the lactate decrease was seen in treated cells. Not clear that you can directly compare them.

The point of the experiments shown in Fig. 3a and Fig. S5a is that, despite being a glycolytic mutant, $\Delta gpmA$ Salmonella are at least as capable of balancing redox and producing ATP as wild-type cells. These findings motivated us to check respiratory activity, which showed that $\Delta gpmA$ Salmonella consume more oxygen than wild-type controls. The increased respiratory activity recorded in $\Delta gpmA$ Salmonella provides a reasonable mechanism for the apparent ability of this glycolytic mutant to maintain redox balance and ATP synthesis. The text has been modified to more accurately convey our reasoning.

Line 162. What is your model to explain how GpmA would influence your proposed dissipation of ΔpH ?

According to our investigations, $\Delta gpmA$ Salmonella suffer a more profound inhibition of ΔpH upon H₂O₂. Overutilization of the noncoupled NDH2 isoform by $\Delta gpmA$ Salmonella offers a reasonable explanation for the profound loss of ΔpH seen in this strain. This idea has been more clearly presented in the results.

Fig S5. One of your primary conclusions is that H₂O₂ inactivates NDH1. The rate of this in vitro reaction suggests that you are simply oxidizing any thiols in the protein ([https://doi.org/10.1016/S0891-5849\(99\)00051-9](https://doi.org/10.1016/S0891-5849(99)00051-9)). More importantly, Imlay and colleagues have shown that NDH1 is NOT damaged in vivo by H₂O₂ even at 5 mM ([doi:10.2007/1074/jbc.M607646200](https://doi.org/10.2007/1074/jbc.M607646200)).

We agree with the reviewer that H₂O₂ is not likely to directly damage NDH1. However, other oxidants can inhibit NDH1. We have modified the model to consider that the inhibition seen under our experimental conditions is probably indirect. In addition to chemical inhibition of NDH1 by ROS, we would like to point out that our investigations show that oxidative stress represses *nuo* gene expression, likely contributing to the overall diminution of NDH1 activity in live cells. This idea has now been developed in more detail in the Results and Discussion.

Your interpretation of the DsbA data seems backwards. Most importantly, the previous evidence shows that the role of DsbA is to introduce disulfide bonds into periplasmic proteins. DsbC and G act to repair oxidized thiols.

We are aware that the canonical role of DsbA is to introduce disulfide bonds in periplasmic proteins. As such, the catalytic cysteine residues in DsbA must be kept oxidized by the membrane bound DsbB, whose enzymatic activity relies on the ETC. In *E. coli*, a small fraction of DsbA has consistently been found in its reduced state. Investigators have reasoned that the fraction of reduced DsbA might be a technical artifact. We were surprised to find that over 50% of DsbA is reduced in Salmonella. Treatment of Salmonella with the uncoupler CCCP resulted in complete oxidation of DsbA, strongly suggesting that technical issues cannot easily explain the large fraction of reduced DsbA recorded in Salmonella. Moreover, the predominantly oxidized DsbA in CCCP-treated Salmonella coincides with poor protein folding and increased

susceptibility to H₂O₂. The genetically modified DsbA^{trxA} variant is also overwhelmingly oxidized. Contrary to what could be predicted from current models, Salmonella expressing this predominantly oxidized variant is defective at folding a periplasmic target, is hypersusceptible to H₂O₂, and is attenuated in NOX2 sufficient mice. Together, our investigations indicate that Salmonella benefits from retaining a sizable fraction of DsbA in its reduced state. One possibility, presented in the original version, is that reduced DsbA plays previously unsuspected functions as a reductase; however, other scenarios cannot yet be ruled out. It is possible that overoxidation of DsbA through high electron flow in the ETC stresses cell envelope thiol-disulfide oxidoreductases. This information has now been added to the Discussion. The model presented in Fig. 5 has been modified to more clearly convey the idea that redox balancing in glycolysis diminishes electron flow through ETC, not only preserving thiol-disulfide oxidoreductases in the cell envelope but also protecting the bacterial cell from oxidative stress.

Reviewer #2 (Remarks to the Author):

In this manuscript, the authors set out to answer the long-debated question of how NADPH oxidase kills microbes. The authors present sound data supporting the hypothesis that NADPH oxidase collapses the ΔpH of intracellular Salmonella and synergizes with hydrogen peroxide to maximize antimicrobial activity. Additionally, the authors show that a microbial adaptation to resist the respiratory burst lies in the shift from ETC to glycolysis.

The experiments are presented logically and support the author's hypothesis in a -given the complex nature of the subject- relatively easy to understand way. The experiments were performed rigorously and to high scientific standard. The authors employ mutants in glycolysis enzymes in their study and pleiotropic effects could have easily masked relevant findings. However, the authors present numerous highly relevant controls for their experiments.

The findings of the study are novel and should interest a broad readership.

We appreciate the reviewer's comments.

Specific comments:

Statistical analyses should be provided for: mouse experiments, FigS3 (according to M&M analysis was done by ANOVA, but is not indicated in the figure), 3d (no statistical difference to $\Delta\text{gpmA ndh}$?).

Statistical analysis has been included for mouse experiments. Deletion of \$\text{ndh}\$ significantly increases resistance of \$\Delta\text{gpmA}\$ Salmonella. The statistical analysis is shown in revised Fig. 3d.

M&M describes complementation of mutant strains. For most experiments, multiple mutants were analyzed that showed similar effects, making off-target effects unlikely. However, key experiments with complemented strains should be presented.

As requested, we now show complementation of \$\text{gpmA}\$ in other key experiments, including competition assays, oxygen consumption, and intracellular pH.

If in 4d and S7d representative blots are shown, then more should be available to show the average of densitometric analyses.

As requested, the average of the densitometric analyses is shown in Figs. 4d and S7d.

Admittedly, I could not follow rationale or conclusions for Fig. 4g-i. A few additional sentences might clarify this.

As requested, we have added more explanations for Fig. 4g-i.

Y axis are frequently labeled in a rather unconventional way that requires conversion in order to interpret. Readers will be familiar with $OD_{600}=1$; presenting it as 10^0 is unnecessarily confusing. I do not see any reason for presenting survival logarithmically. And again, presenting it as 10^{-1} to 10^2 is confusing to the reader.

We believe the reviewer is referring to the results representing the killing of Salmonella by H_2O_2 . Due to the magnitude of killing (up to 100-fold in some of the assays), it would be difficult to see the differences when shown in linear representation. To address the criticism raised by the reviewer, we have represented the data in antilog scale.

Fig. 1c: The reduced survival with succinate should be explained.

Succinate increased resistance of $\Delta gpmA$ Salmonella to H_2O_2 , but did not reach the levels of wild-type controls. According to our model, it is possible that the addition of electrons to the ETC during utilization of succinate may have contributed to the increased killing of $\Delta gpmA$ Salmonella compared to wild-type controls grown in the presence of succinate. This information has been added to the revised manuscript.

Fig 2A could additionally serve as helpful reminder as to where in the pathway the enzymes relevant for Fig. 1 and other figures are active. Please add the few enzymes that are relevant for the paper. This should be possible without being too distracting for the findings of Fig. 2A and could facilitate the reader's understanding.

As requested, the names of the genes encoding relevant enzymes have been added to Fig. 2A.

Could the authors provide the reasoning behind different concentrations of H_2O_2 used? 400uM for most assays, 2.5mM for metabolomics. After 2h with 400uM only 0.2% of $gpmA$ is alive, after 30min with 6x this concentration I would assume that cell viability is affected, yet M&M states that no viability loss was seen?

Killing by H_2O_2 is very much dependent on cell densities: the denser the culture, less efficiently peroxide kills. The text is correct: after 30 min of treatment with 2.5 mM H_2O_2 the viability of dense cultures was not affected, as the OD of treated cultures was ~ 0.6 (amounting to $\sim 7-8 \times 10^8$ CFUs/ml). By 2h after treatment with 2.5 mM H_2O_2 , viability dropped to $\sim 1-1.2 \times 10^8$ (a ~ 7 fold decrease). In comparison, 2 h after the addition of 400 μM H_2O_2 , 2×10^5 CFUs of $\Delta gpmA$ Salmonella drop to $\sim 10^3$ CFUs, about a 200-fold reduction. This information has been clarified in the revised Methods and legend of Fig. 1.

Fig. S3.: Check descriptions in text: succinate does not seem to be lower in untreated $gpmA$ compared to WT. Also 3-phosphoglycerate (as expected) higher in $gpmA$.

The text has been corrected accordingly.

Check color and legend for Fig. 2d. The red should be purple/blue as well?

Sorry for the error. This is the ppc $ackA$ pta strain in C57BL/6 mice. The symbols have been fixed.

3f: please describe the effect of addition of benzoate. It is mentioned in the M&M, but as appears in a main figure, it should be explained.

Usage of benzoate as a protonophore has been clarified in the Methods.

4a: why is gpmA alkaline phosphatase at 0mM H₂O₂ so much lower with CCCP than WT?

The assay was done by growing overnight cultures of phoA-expressing wild-type and Δ gpmA Salmonella in low phosphate MOPS-GLC media to about OD₆₀₀ of 0.4. The cells were then normalized to OD₆₀₀ of 0.3 and treated for 30 min with H₂O₂, CCCP, or a combination of H₂O₂ and CCCP. As folding of periplasmic proteins by the Dsb system relies on the PMF, the depolarization induced by CCCP likely explains the drop in alkaline phosphatase. To help the reader better interpret the data, we have edited the Results, and Methods.

Reviewer #3 (Remarks to the Author):

This comprehensive study of Salmonella typhimurium by the authors provides compelling evidence for the gene gpmA, which encodes for glycerophosphomutase, having a key antioxidant role by facilitating a switch to glycolysis during oxidative stress. The finding is demonstrated to be relevant to Salmonella pathogenesis using a mouse infection model. It is generally understood that glycolysis benefits cells under oxidative stress, particularly in cancer. The novel finding of this study is identifying gpmA as a key player in the metabolic switch and the discovery of its role in oxidative stress protection and pathogenesis. Overall, this is a very thoughtful and significant contribution that is of high importance in the field of metabolism and cellular stress response. The conclusions are supported by convincing and high quality data and the manuscript significantly advances knowledge of metabolic reprogramming during stress.

Thank you very much for your kind comments.

Some suggestions/corrections.

Line 494, Figure 1C. What Salmonella strain is being used? Please confirm it is the gpmA mutant. This also needs to be made clear in Figure 3e.

Fig. 1C compares susceptibility of wild-type and Δ gpmA Salmonella to H₂O₂ when grown in MOPS minimal media with the indicated carbon sources. To clarify this information, symbols have been added to the Fig. 1C.

Line 66, deferoxamine did not protect the hydrogen peroxide killing of the gpmA mutant. Were other antioxidants tested as well? e.g, N-acetylcysteine

We also tried dipyrindyl with similar results. The data have been added to revised Fig. S1c.

3. In Figure 2, define ldhA (D-lactate dehydrogenase gene)

The ldhA gene has now been added to Fig. 2a.

What are the red squares in Figure 2d? gpmA mutant strain?

Sorry for the error. This is the ppc ackA pta in C57BL/6 mice. The symbols have been fixed.

Figure 4d, Need to indicate whether hydrogen peroxide is present or not in lane 3 of the figure.

The negative symbol has been added to the Fig. 4d.

Figure 4J-The reaction arrow for DsbB is in the wrong direction relative to DsbA.

Given the concerns raised by reviewer #1, we have simplified the model presented in Fig. 5.

Consider using the enzyme nomenclature of phosphoglycerate mutase instead of glycerophosphomutase.

As suggested, the change has been made.

Add a comment about whether there are significant changes in the GSH/GSSG ratio.

The metabolomics showed no differences in GSH or GSSG between wild-type and Δ gpmA Salmonella. This information has been added in the revised manuscript (Fig. S4).

Provide more of an explanation for the higher ATP levels found in the gpmA mutant cells relative to WT (line 136 and Fig. S5a). Also, what is the impact of hydrogen peroxide treatment on the ATP levels in the gpmA mutant cells relative to the WT strain?

The increased ATP levels seen in Δ gpmA Salmonella likely reflect the higher respiratory activity of this mutant. This explanation has been added to the revised manuscript.

Treatment of wild-type or Δ gpmA Salmonella with H_2O_2 lowered the concentration of ATP. This new experiment has been added to the revised manuscript (Fig. S6a).

REVIEWERS' COMMENTS:

Reviewer #1 (Remarks to the Author):

Chakraborty et al characterize the sensitivity of a mutant lacking the metal-independent glycerophosphomutase GpmA to hydrogen peroxide and, by analogy, the oxidative burst of host immune cells. This is a much improved manuscript. Although the overall mechanism remains unclear, the explanations of individual results now make the paper easier to follow for the reader.

I would suggest that the authors go through the figures and provide additional labeling, eg Fig 3d is nicely labeled "+400 μ M H₂O₂, 2h", but Fig 1b, etc are not. Fig 3h etc should be labeled as macrophages rather than just the mouse source of the macrophages. Try to make it so the reader doesn't have to read the legend to understand the results.

Reviewer #2 (Remarks to the Author):

The authors carefully addressed all my concerns. The requested experiments with complemented mutants show the expected phenotype. Careful rephrasing and additional explanations as requested by another reviewer, helped to clarify and strengthen the manuscript.

REVIEWERS' COMMENTS:

Reviewer #1 (Remarks to the Author):

Chakraborty et al characterize the sensitivity of a mutant lacking the metal-independent glycerophosphomutase GpmA to hydrogen peroxide and, by analogy, the oxidative burst of host immune cells. This is a much improved manuscript. Although the overall mechanism remains unclear, the explanations of individual results now make the paper easier to follow for the reader.

Thank you for your support.

I would suggest that the authors go through the figures and provide additional labeling, eg Fig 3d is nicely labeled "+400 uM H₂O₂, 2h", but Fig 1b, etc are not. Fig 3h etc should be labeled as macrophages rather than just the mouse source of the macrophages. Try to make it so the reader doesn't have to read the legend to understand the results.

The suggested changes have been added to the figures.

Reviewer #2 (Remarks to the Author):

The authors carefully addressed all my concerns. The requested experiments with complemented mutants show the expected phenotype. Careful rephrasing and additional explanations as requested by another reviewer, helped to clarify and strengthen the manuscript.

Thank you for your support.